# Energizing Workplace Dynamics: Exploring the Nexus of Relational Energy, Humor, and PsyCap for Enhanced Engagement and Performance

**DOI:** 10.3390/bs14010023

**Published:** 2023-12-27

**Authors:** Medina Braha, Ahu Tuğba Karabulut

**Affiliations:** 1International Business College Mitrovica, 40000 Mitrovica, Kosovo; 2Business Faculty, Istanbul Commerce University, Istanbul 34445, Turkey; tkarabulut@ticaret.edu.tr

**Keywords:** relational energy, humor, psychological capital, positive leadership, management, organizational behavior, human resource management, interpersonal communication, performance, COVID-19

## Abstract

This study delves into the dynamics of relational energy (RE) within an organizational context, examining some of its antecedents and decedents. Specifically, it investigates the influence of psychological capital (PsyCap) and humor on RE, and, subsequently, the latter’s impact on job performance (JB) mediated by job engagement (JE). A research model based on structural equation modeling carried out with 481 employees in private service industries demonstrates several key relationships. It reveals that both PsyCap and affiliative humor positively affect RE, while aggressive humor exerts a negative influence. Furthermore, RE shows a positive association with JE and JP, with JE serving as a mediator. To the authors’ knowledge, this is the first study to present an integrated model encompassing this exact combination of influencers and consequences of RE, as well as the first to be investigated within the Western Balkans cultural context. Therefore, it represents a novel approach. Additionally, the research addresses crucial questions regarding the existence and strategic significance of RE within organizational interactions. The findings offer valuable insights for organizations seeking to enhance employee engagement, performance, and wellbeing—even during health crises such as COVID-19—by fostering RE. This study advances the understanding of RE in organizational settings and provides a foundation for future research in this domain.

## 1. Introduction

Positive human energy is endorsed as a source of high-quality relationships, which in turn, foster individual and organizational excellence [1]. Relational energy is “a heightened level of psychological resourcefulness generated from interpersonal interactions that enhances one’s capacity to do work” [2] (p. 37) and is claimed to be synonymous with positive energy within human interactions [3].

The essential objective of this work is to determine ways of enhancing relational energy, deriving from a dyadic interaction, and its associated benefits within organizational life. This aim is embarked upon through establishing the correlation of PsyCap and humor with relational energy, i.e., the former are postulated as impactors of the latter. In addition, the impact of relational energy on job engagement and job performance is examined based on Owens et al. [2].

Considering it is still in the developing stage, the field of relational energy needs many additional theoretical and empirical explorations. Hence, the current study intends to contribute to reducing that research gap. Furthermore, this study stands as one of the few studies delving into the antecedents of relational energy. It also distinguishes itself by not solely focusing on a leader-member approach—as a large number of relational energy researchers do—but by generalizing its scope to encompass horizontal interactions within the organization. Moreover, to the authors’ knowledge, no prior works have established this precise correlation among all the variables as presented in the current study. Finally, this study’s novelty entails also presenting a model that simultaneously examines both factors contributing to and outcomes resulting from relational energy, a unique contribution in the existing literature.

The correlation is theoretically reasoned building upon positive organizational scholarship (POS) [4], positive organizational behavior (POB) [5], social contagion theory [6], interaction ritual theory [7,8], and conservation of resources theory [9]. Following the main objective, this study aims to answer the following three research questions:Is there a presence of relational energy within the interpersonal communication between employees of a unit or organization?Are there means available for enhancing relational energy and its associated benefits in organizational settings?Are there substantial benefits from relational energy that deem it of strategic importance for the organization’s management to consider?

This study responds simultaneously to the first and second research questions by exploring the role of PsyCap and humor as potential tools for amplifying the gains derived from relational energy within a team, unit, or organization. Establishing potential correlations among humor, PsyCap, and relational energy tends to confirm the existence of relational energy (research question 1) and the means for its advancement (research question 2). Finally, the study will examine the influence of relational energy on both job engagement and job performance, illustrating its significant paybacks that deserve consideration from the organization’s management. (research question 3).

The current study first provides a theoretical background and elaboration in support of the proposed model, derived originally from Braha [10,11]. Consequently, this study tests the argued model and discusses the obtained findings. Finally, this paper concludes by suggesting its contributions, limitations, and recommendations for future research.

## 2. Literature Review and Research Hypotheses

### 2.1. Relational Energy

Positive deviance within the work context is chiefly studied by POS and POB, both originating from positive psychology [12]. The basic idea of positive deviance is how to move from good or normality to excellence. Examples outside of work settings include the health conditions of an athlete and the intelligence level of a genius, while within organizational settings it includes the exceptional performance of an employee [3]. This research attempts to synergize constructs from both disciplines, POS and POB, so as to boost relational energy and its associated benefits.

Unlike physical, mental, and emotional energies that exhaust when used, relational energy increases the more it is utilized [3,13]. Research on relational energy builds upon the studies of energizers and de-energizers [14,15]. However, these earlier studies do not explicitly address the concept of relational energy as explored in the current research. McDaniel [16] takes the initial step in developing and conceptually defining relational energy. She makes a distinction between emotional energy and relational energy, with the latter being identified as a resource-based process of exchange. Furthermore, her empirical investigation shows that organizational members are aware of and can easily discuss the dynamics of energy in the workplace, with culture seemingly playing a minimal role in how relational energy functions within an organization. Accordingly, the author suggests that relational energy might be regarded as a universal phenomenon. McDaniel’s [16] scale aimed to measure the exchange of relational energy from the energy giver’s perspective. However, Owens et al. [2] argue that understanding the perception of the energy recipient is essential for a comprehensive grasp of the energizing process. Subsequently, they pioneer the receiver-centric perspective on relational energy by establishing and validating a 5-item scale for measuring it. In developing this construct, they draw upon three scientific theories: interaction ritual theory, conservation of resources theory, and social contagion theory. Through their theoretical and empirical analysis, the authors differentiate the notion of relational energy from related constructs of social support, relational identification, productive energy, emotional energy, and leader–member exchange (LMX), thereby instituting it as a psychometrically reliable, valid, and robust concept. As a result, the field of relational energy within the broader realm of human and organizational energy studies could be considered as being still in its developing stage.

### 2.2. Relational Energy and Psychological Capital

Several scholars suggest that psychological capital (PsyCap) goes beyond human and social capital since it enables people to progress from the actual self to the potential self/collective selves [17,18]. PsyCap appears to positively affect several dimensions such as life satisfaction [19], employee basic need satisfaction [20], innovation [21], creativity [22], organizational citizenship, job performance, psychological wellbeing, and organizational commitment [23], as well as reduce turnover [24], anxiety, and cynicism [23]. Moreover, increased levels of PsyCap seem significantly beneficial in COVID-19 lockdown working conditions [25,26,27,28,29,30,31].

Furthermore, numerous studies show team or collective PsyCap acting as an antecedent, mediator, or moderator of safety performance [32]. Similarly, Newman et al. [33] summarize various implications of team PsyCap on team-level outcomes (performance, satisfaction, engagement, creativity/innovation) and organizational PsyCap on organizational-level outcomes (firm performance, innovation, growth). Collective PsyCap is also found to partially mediate the correlation between shared leadership and both creativity and organizational commitment [34].

The correlation of PsyCap with relational energy is illustrated by PsyCap-related qualities that energizers have as compared to de-energizers. Energizers transmit hope to those they interact with, see realistic new possibilities [15], stand optimistic [35], and follow through [3]. Conversely, de-energizers often are critical [3] and focus on obstacles [15]. In addition, people higher on PsyCap show a higher relational energy state and a contagious influence since they boost others’ optimism and hope [25] and bring positivity into their private and workplace social relationships [36]. In view of that, building upon contagion theory and interaction ritual theory, and drawing from PsyCap features found in energizers, it is argued that people higher in PsyCap tend to energize more, i.e., generate higher relational energy.

**Hypothesis** **1.**
*There is a significant association between PsyCap and relational energy.*


### 2.3. Relational Energy and Humor

Humor is also found to produce valuable physical, social, and organizational effects. Examples of such positive impacts are related to, for instance, blood pressure [37], stress [38], effectiveness [39,40], innovation [41,42], trust [43], burnout, work withdrawal [44], persistent behavior [45], job satisfaction, organizational pride, affective commitment [46], engagement [47], social cohesion [41], creativity communication, enthusiasm, and brightened and more enduring workplace [48]. Humor too seems to be helpful for employees and organizations during the COVID-19 setbacks and aftermaths [49,50,51].

The association of humor with relational energy made in the current research is interpreted as people who receive higher ratings for positive humor are expected to display a higher relational energy state. Humor is also studied by psychology outlook [12] and POS [52]. Greater utilization of positive humor positively reflects on passing interaction [52], social relationships [53], high-quality connections [1], positive energy among the user and receiver of humor [3,15], improved communication [48], reduced tension [54], social cohesion, solidarity, and rapport [41]. Furthermore, Cheng and Wang [45] argue that humor can produce amusement, a particular type of positive emotion, and can help people replenish work-related resources. Recently, Simione and Gnagnarella [55] found that humor enhances positive emotional states. On the other hand, positive emotions are very frequently revealed in relational energy [16]. Considering relational energy increases the more it is used and diminishes in de-energizing interpersonal communication, and that humor, as per the above explanations, can create positive emotions and positive energy, the work-related resources referred by Cheng and Wang [45] can be argued to include relational energy as well. On this basis, building upon interaction rituals theory and conservation of resources theory, and referring to the construct of positive emotions, it is suggested that people who utilize more positive humor tend to energize more.

It is imperative to emphasize that not all humor types represent pleasant interactions. To demonstrate that, the research is based on the humor styles of Martin et al. [56], i.e., self-defeating, self-enhancing, affiliative, and aggressive, and the associated Humor Styles Questionnaire (HSQ). Numerous studies offer evidence that adaptive humor styles (affiliated and self-enhancing) of HSQ are positively associated with psychological wellbeing, facilitating relationships, and reducing interpersonal conflicts, whereas maladaptive humor styles (self-defeating and aggressive) of HSQ show a negative correlation [57]. Building upon this logic, the former are considered positive humor and are expected to be positively related to relational energy, whereas the latter are considered negative humor and are expected to be negatively related to relational energy.

**Hypothesis** **2.**
*There is a significant association between humor and relational energy.*


### 2.4. Relational Energy as a Source for Employee and Organizational Wellbeing and Performance

Having identified two antecedents of relational energy, it is crucial to also show employee and organizational benefits that derive from it. After all, the level of relational energy without organizational repercussions, though probably valuable in personal relationships, seems not a relevant cause to be considered within organizational and management research. POS’s credibility depends partly on its capability to show that organizational positivity is correlated with organizational performance; otherwise, organizations most likely will not allocate resources for the purposes of developing and applying positive practices [52]. Hence, the current study’s second part turns to the benefits of relational energy for organizations by focusing on the effect of relational energy on job performance through the mediating role of job engagement.

In addition to their own, energizers seem to also progress the performance of others who are linked to them or interact with them [3,13,35]. Human energy is socially contagious [2,3,14,16,58], while organizational energy is augmented from individual energies and fuels readiness to change, innovation, and productivity [59]. As such, positive energy transmitted from one person to another results in healthy work relationships among people of a team, unit, or organization that lead to improvements in individual and organizational thriving [60], mental sharpness, learning pace, memory, post-surgery recovery, experiences of depression, sickness, and discomfort [61], immune, cardiovascular, and hormonal system [62] performance, as well as engagement [1,2,14,15], creativity, motivation, uplifting, elevation, vitality [3,13,35], and knowledge transfer [63]. Very importantly, high-quality connections originating from positive energy generate further positive energy within interpersonal communication, resulting as such into a powerful virtuous cycle of energy-HQC generation and transmission [1]. Notably, positive outcomes derive not as much from what people gain from those relationships, but also from what they invest in them [64,65,66].

Relational energy is found to produce manifolds of direct or mediating positive outcomes such as job engagement, job performance [2,67,68,69], work passion transmission [70], interpersonal citizenship [71], high-quality mentoring relationship [72], deep acting [73], customer service engagement behavior [74], relationship quality [75], perceived relational climate [76], and reduction of consequences of work–family conflict [77]. In addition, relational energy can moderate the disadvantageous effect of emotional labor and improve cognitive flexibility [78] as well as moderate the detrimental influence of digital connectivity during COVID-19 lockdowns [79].

The reasoning of Owens et al.’s [2] association of relational energy with job performance is to be found in earlier works [14,59] showing energizers significantly impacting performance. Likewise, the mediating role of job engagement to job performance is also visible in previous research [80,81]. Later research was also carried out which specifically investigated the effect of relational energy on job engagement and/or job performance [67,68,69,77,79], the majority of them utilizing the instrument developed by Owens et al. [2].

In summary, organizations are unlikely to prioritize the development of relational energy unless they anticipate tangible benefits such as enhanced employee performance, engagement, or wellbeing. Considering the supporting evidence for the positive effects of relational energy and energizing relationships, it is worthwhile for organizations to focus on fostering and sustaining this positive energy among their employees, teams, units, and throughout the organization.

Drawing from the above, the following 4 hypotheses are postulated.

**Hypothesis** **3.**
*There is a significant association between relational energy and job engagement.*


**Hypothesis** **4.**
*There is a significant association between relational energy and job performance.*


**Hypothesis** **5.**
*There is a significant association between job engagement and job performance.*


**Hypothesis** **6.**
*There is a significant mediating role of job engagement in the association between relational energy and job performance.*


## 3. Research Method

### 3.1. Research Model

This research analyzes relational energy from two different perspectives—what affects it and what impact it has on organizational life—by suggesting an integrated model of some antecedents and descendants of relational energy. The utilized model consists of two main parts with relational energy being the center of it. The first part of the model explores PsyCap and humor as drivers of relational energy, while the second part investigates the effect of relational energy on job performance and job engagement, with job engagement being the mediator of the RE-JP relationship.

Bearing in mind all the above discussed, the model is postulated as in Figure 1.

### 3.2. Research Sample and Measurement Instruments

The sample consists of private sector service companies headquartered in the region of Prishtina, Kosova where data collection was conducted during the period of 2017–2018. Service industries are usually characterized by people working together and every so often some are placed in the same office premises. From the Kosovo Agency of Statistics [82] company size classification—micro (0–9 employees), small (10–49 employees), medium (50–249 employees), and large companies (250+ employees)—the latter two types were chosen. Considering relational energy is measured by assessing co-workers, micro and small companies were excluded in order to avoid bias since they merely consist of family or friends entering into business together who had known each other for many years. In contrast, medium and large companies represent more diversity in terms of staff. These were considered suitable circumstances for measuring relational energy more objectively. Moreover, Luthans & Youssef-Morgan [18] determine that PsyCap is more robust with outcomes in the service sector. Service industries were included as per the Kosovo Agency of Statistics [83] categorization of economic activities which is rooted in ISIC Rev.4 [84] and NACE Rev. 2 [85], namely from category G to S.

Respondents are from both genders, with a random selection in that regard. In the same logic, the participants’ age group, education level, tenure, and hierarchical position consist of a range of different backgrounds. The important part was for the respondents to rate the last coworker at the same hierarchal level (a peer, not a superior or direct subordinate) with whom they had worked closely for a duration not shorter than 3 months. As such, the sample did not consist of heads of the company (CEO) since they have no peers as per the above description.

Considering the study is cross-sectional, respondents received only 1 questionnaire. In the initial section of the questionnaire, respondents assessed the relational energy, PsyCap, and humor of the coworker. The subsequent section included a self-assessment of respondents’ own job engagement and job performance.

Relational energy was measured through the 5-item scale developed and validated by Owens et al. [2]. PsyCap was measured through the 12-item scale developed and validated by Avey et al. [17] from the original 24-item PsyCap Questionnaire (PSQ) developed and validated by Luthans et al. [86]. The peer-rating version was obtained from Mind Garden (http://www.mindgarden.com/136-psychological-capital-questionnaire accessed on 16 January 2017). Humor was measured through the Humor Scales Questionnaire (HSQ) developed and validated by Martin et al. [56]. A short version of 20 items (from the original 32 items) and a peer-rating version of the original scale were sent through e-mail from R. A. Martin himself by request of the current study’s first author. For purposes of this study, the peer-rating short version was developed by the author. Job engagement was measured through a self-rating scale with 9 items developed and validated by Schaufeli et al. [87]. Finally, job performance was assessed by a combination of two self-rating scales. The first scale uses 13 out of 18 items (contextual performance and task performance, excluding counterproductive work behavior) developed by Koopmans et al. [88] and validated by Koopmans et al. [89]. The second scale was adapted from Hwang [90] who makes use of the 4-item scale of Tsui et al. [91] with 2 added items (6 items in total). This scale evaluates performance from the standpoint of its quality, quantity, and efficiency. All instruments are in English. In order to adjust the respondents’ native language, all items were translated into Albanian by the first author in consultation with other researchers.

## 4. Results

### 4.1. Analytical Approach

All submitted questionnaires were checked for their completeness and the quality of data. Out of 549 completed questionnaires 481 were proved to be valid and the overall response rate was 87%. The threshold of 2% was used for missing responses and removing questionnaires from the data set.

Structural Equation Modeling was considered appropriate, while Partial Least Square Structural Equation Modeling (PLS-SEM) with SmartPLS3 was employed to examine the research model. A sample of more than 200 respondents is needed to validate the effectiveness of Structural Equation Modeling [92]. According to this criterion, our study meets the validity related to sample requirements.

Internal consistency and reliability analysis for LIKERT scale variables was performed using Cronbach’s Alpha coefficient. According to Nunnally [93], the variables in each scale have a high degree of reliability and are positively related to each other if Chronbach’s Alpha is at least 0.7.

After the achievement of internal consistency and reliability, convergent and discriminant validity were aimed through Confirmatory Factor Analysis (CFA). Assessment of convergent validity was tested with Composite Reliability (CR) and the Average Variance Extracted (AVE). The acceptable level values for the latent constructs are CFA > 0.7, CR > 0.7, and AVE > 0.5 [94]. As a result, items with a loading factor less than 0.7 were excluded from the model. All 5 items for the RE had a loading factor greater than 0.7; therefore, all of them were included in the model. As per the other constructs, the total number of items included in the model is 3 for affiliated humor, 3 for self-enhancing humor, 3 for aggressive humor, 2 for self-defeating humor, 7 for PsyCap, 7 for job engagement, and 4 for job performance. The excluded items from HSQ are items 8, 10 (affiliated humor), 13 (self-enhancing humor), 2, 6, 16 (aggressive humor), 3, 9, and 11 (self-defeating humor). Items 5, 9, 10, 11, and 12 were omitted from PsyCap, items 8 and 9 from job engagement, and items 1–8 and 13–19 from job performance. It is noteworthy that only the contextual dimension of job performance remained measurable after the item omissions.

Discriminant validity was tested with Heterotrait–Monotrait (HTMT) matrix, the value of which should be below 0.90 [94,95]. After the CFA analysis, the research model was assessed by calculating the sum of variance on relational energy to job performance explained by psychological capital, affiliated humor, self-enhancing humor, aggressive humor, self-defeating humor, and job engagement. Standardized Root Mean Square Residual (SRMR) was utilized to achieve model fit, whereby the values below 0.1 are acceptable for model validation [95,96]. Mediation effect of job engagement between relational energy and job performance was estimated as suggested by Baron & Kenny [97]. The level of mediation effect was assessed with the variance accounted for (VAF), whereby a value higher than 80% indicates full mediation; a value in the range of interval 20–80% indicates partial mediation, and a value smaller than 20% shows that there is no mediation effect [98].

Lastly, the exploratory model of antecedents and descendants of relational energy was also controlled for demographic factors (age, gender, tenure, position, etc.). None of the demographics used as control variables appeared to be significant.

### 4.2. Descriptive Statistics, Reliability, Validity, Model Fit, and Hypotheses Testing

Based on the results presented in Table 1, it can be seen that relational energy has the highest mean, while aggressive humor has the lowest, whereas the standard deviation is highest for self-defeating humor and lowest for job performance.

Table 2 presents correlation coefficients between the constructs included in the model. According to the obtained results, affiliated humor, self-enhancing, and self-defeating humor are statistically significant and positively correlated with each other, while physiological capital is positively and statistically significantly correlated to affiliated humor and self-enhancing humor.

Each construct in the model has Cronbach’s Alpha greater than the minimum threshold of 0.7, as presented in Table 3, which is considered to be reliable according to Nunnally [93]. The construct of affiliated humor (0.964) and relational energy (0.927) have the highest value of Cronbach’s Alpha.

As per the confirmatory factor analysis, the indicator loadings in Table 4 show good indicator reliability, as all loadings are larger than the threshold. Composite reliability for each construct is higher than 0.7 and the AVE values are all above 0.5.

The model also appears to be valid in terms of discriminant validity as all values of the HTMT matrix for the latent constructs are below the threshold of 0.90, as presented in Table 5.

Significant evidence was obtained for Baron & Kenny’s [97] conditions to be met. Findings show that relational energy seems to significantly affect job performance and impact job engagement (the mediator), and relational energy and job engagement significantly influence job performance.

Figure 2 shows the relationship of the independent variable predicting the independent variable through the mediator. Psychological capital results in a strong and positive impact on relational energy (β_PC-RE_ = 0.405; t = 8.617; *p* = 0.000), confirming Hypothesis 1. Affiliated humor is positively related to relational energy, but the result does not appear statistically significant (β_AFFH-RE_ = 0.076; t = 1.658; *p* = 0.097). Self-defeating humor is also positively correlated to relational energy, but this correlation, too, does not seem statistically significant (β_SDH-RE_ = 0.041; t = 1.068; *p* = 0.286). These two results do not provide sufficient evidence in support of Hypothesis 2. However, self-enhancing humor results positively related and statistically impacting relational energy (β_SEH-RE_ = 0.220; t = 5.086; *p* = 0.000), whereas, aggressive humor, on the other hand, is found to be statistically significant and negatively impacting relational energy (β_AGH-RE_ = −0.101; t = 2.374; *p* = 0.018). These results provide sufficient evidence in support of Hypothesis 2. Relational energy appears to positively impact job performance (β_RE-JP_ = 0.092; t = 2.125; *p* = 0.034) and job engagement (β_RE-JE_ = 0.282; t = 6.705; *p* = 0.000), providing evidence in support of Hypotheses 3 and 4, respectively. Finally, the results show job engagement having a significant mediating influence on relation energy’s impact on job performance (β_RE-JE-JP_ = 0.110; t = 4.583; *p* = 0.000), confirming Hypothesis 6.

Once confirmation of the mediating role of job engagement was obtained, the strength of this mediation was examined. The results are displayed in Table 6.

Following Hair et al.’s [95] recommendation, the computed value of the current model indicates that job engagement serves as a full mediator in the correlation between relational energy and job performance.

## 5. Discussion

The conducted research finds empirical support for all relationships analyzed with the exception of affiliative and self-defeating humor. Employees with higher levels of PsyCap tend to energize their colleagues more. Even though, to the authors’ knowledge, it is the first time this exact correlation has been tested, still, associations with related works can be made. PsyCap-related features and a contagious effect are found in energizers: following through [3], being optimistic [35], creating hope in others (followers) [15], positively affecting private and workplace social relationships [3], creating positive emotions [99], and increased followers’ optimism and hope [25]. On the other hand, de-energizers appear to be frequently critical [3] and see primarily roadblocks [15]. These examples can explain the resulting support for Hypothesis 1.

Humor’s influence on relational energy was partly supported, namely only self-enhancing and aggressive humor styles resulted in statistical significance. Accordingly, those who utilize more self-enhancing humor tend to energize others more, whereas people using aggressive humor tend to de-energize others, i.e., diminish the relational energy. Humor’s reliability problem could be associated with potentially biased responses as a result of the questions’ order in the instrument or the chance that reversed items—present only within the humor scale—were not completely understood. Recall that humor is the only variable that contains reverse items.

Some emerging research relates humor with relational energy, where the latter serves as a mediator. For instance, Yang et al. [100] find that leader humor positively impacts employee creativity through the mediating role of relational energy; Cheng et al. [101] find that leader humor is positively associated with customer-oriented organizational citizenship behavior, while relational energy mediates this influence; and Zhang et al. [102] find leader humor impacting employee bootlegging through the influence of relational energy. On the other hand, Huang et al. [103] show the negative correlation of employee humor with leader abusive supervision through the role of leader relational energy.

Same as with PsyCap, affiliations with other works can also be made. Positive humor creates high-quality connections [1], both horizontally and vertically [52], facilitates social relationships [53], raises positivity among user and receiver of humor [104], creates the positive emotion of amusement [45], enhances employee wellbeing through positive affect [105], is able to reload work-related depleted resources [45], improves communication, increases creativity and enthusiasm, brightening the workplace, and sometimes making it more enduring [48], advances solidarity and social cohesion, building rapport and emphasizing collegiality [41], and reduces tension [54]. While positive (adaptive) humor (affiliated and self-enhancing) positively influences psychological wellbeing, the facilitation of relationships, and the reduction of interpersonal conflicts, negative (maladaptive) humor (self-defeating and aggressive), on the other hand, does the opposite [57]. People dominated by self-enhancing and affiliative humor seem to experience less hopelessness and stress related to the COVID-19 pandemic; consequently, they engage in more protective behaviors, whereas those led by self-defeating and aggressive humor experience the opposite [49]. Comparably, self-enhancing and affiliative humor positively influence emotional labor, while self-defeating and aggressive humor show a negative impact [106]. Affiliative humor results in being negatively correlated with intercultural communication apprehension [107] and attachment anxiety [108] since it seems to produce a sense of security in interpersonal communication. Further, Yaprak et al. [109] observed negative correlations between aggressive humor and challenge and self-commitment (two sub-dimensions of the Psychological Hardiness Scale), while they found positive associations of both self-enhancing humor and affiliative humor with the Psychological Hardiness Scale and the Oxford Happiness Questionnaire Short Form. This positive/negative outcome of different humor styles could illuminate the impact found on relational energy from self-enhancing and aggressive humor, respectively. As expected, a positive influence was also found on affiliative humor, though this association appeared statistically insignificant, whereas the positive effect of self-defeating humor is not in line with the abovementioned explanation; however, this correlation appeared statistically insignificant too. Such correlations can be argued to be in line with the partial supporting evidence for Hypothesis 2.

Relational energy resulted in a positive impact on job engagement and job performance, and job engagement positively correlates to job performance. Thus, results show a positive correlation between relational energy on job performance through the mediating role of job engagement, in line with findings by Owens et al. [2]. Results provide significant evidence for hypotheses 3 to 6 to be accepted. Related works find the mediating influence of amplified follower relational energy in the leader humility–follower task performance correlation [69] and the mediating role of relational energy in the spiritual leadership–employee job performance correlation [67]. Amah & Sese [68] show relational energy enhancing job engagement which is mediated by employee voice and perception of organizational support. Further, Halbesleben & Wheeler [80] and Rich et al. [81] examine the mediating role of job engagement on job performance. Other similar results exist, though not applying the scale developed by Owens et al. [2]. Team energy is found positively linked to team success [15] and energizers are positively related to team and organizational performance [13]. Numerous scholars [2,3,14,16,58] argue that people’s energy is contagious and, depending on whether it is positive or negative, it can positively or adversely impact various dimensions of others’ performance. Ultimately, Chadee et al. [79] demonstrate the benefits of relational energy during the COVID-19 lockdowns. The latter showed adverse consequences of digital connectivity on work behavior due to self-control exhaustion, which, in turn, ends up with work disengagement. Relational energy appears to moderate this detrimental influence of digital connectivity.

## 6. Conclusions

Human energy appears socially contagious and a source of individual and organizational excellence and thriving. Organizational energy represents augmented synergy generated from individual human energy exchanges within a particular team, unit, or organization. As such, it has multifold positive outcomes in terms of interpersonal communication, employee wellbeing, and individual and organizational performance. As a distinct manifestation of human energy, relational energy is considered to increase by use. Additionally, based on this study’s findings, it is also considered to be boosted further by positive humor and PsyCap. Hence, a virtuous cycle of relational energy–high-quality connections is created, i.e., the former develops the latter which, in turn, develops further the former and so on. Many other positive effects of relational energy at work were presented throughout the paper.

This study’s postulated model originates from a combination of positivity streams in organizational studies. It makes multifaceted contributions to the field of relational energy, humor, psychological capital, interpersonal communication, and fostering healthy work relationships, and extends to the broader domains of management and organizational studies. Finally, this is a pioneering examination of the developed model. Hence, there is a whole unexplored area to inspect new evidence on the model’s validity and the proposed relationships.

## 7. Practical Implications

By testing two antecedents, humor and PsyCap, this study targets relational energy’s further development in an organizational context. These two constructs facilitate management’s understanding of how to create, keep, and nurture relational energy within their teams, units, or organization so as to achieve its associated benefits discussed above, particularly so as to serve in augmentation of organizational outcomes such as job engagement and job performance by the proposed integrated model covering all these five variables. The advantages of these new grasps do not necessarily have to be applied only in an organized way. People individually can experience enhanced relational energy by intensifying, on every possible occasion, interactions with those that energize and show signs of positive humor and/or PsyCap as well as by minimalizing the opposite. Furthermore, they can choose to grow their healthy relationships at work by increasing their own relational energy through investments in their PsyCap and positive humor levels. This research raises awareness about the greater effectiveness and efficiency of an organization’s investment (e.g., training, coaching, etc.) in increasing positive humor and/or PsyCap. In addition to their associated individual and organizational outcomes, due to the advancement of employees’ PsyCap and/or humor levels, those investments will probably also lead to greater relational energy among teams and units; thus, there will be further benefits from the same investment. Relatedly, organizations can invest in training employees on how to intensify relational energy, including the two antecedents utilized in this work.

The proposed model appears important in coping with COVID-19 setbacks within organizations. Considering the emerging findings that all these three variables improve COVID-19-related organizational challenges, this particular combination might add up to those results. Furthermore, these insights show the advantages of individual or combined interventions in relational energy, humor, and PsyCap that management can take in order to deal not only with the recent COVID-19 pandemic aftermath, but also with prospective future epidemics, pandemics, or other health crises.

This paper shifts the concentration from leaders being chief generators of relational energy to each employee potentially embracing that function. The current research pioneers the investigation of relational energy in coworker interpersonal communication rather than in a leader–member one. The analysis also extends the existing focus of relational energy’s direct, mediating, or moderating role on certain beneficial outcomes to how relational energy can otherwise be further and differently boosted, above and beyond leadership style and leader behaviors and actions, so that its impact of those outcomes multiplies. Nonetheless, the emerging and yet understudied world of relational energy (and human energy in general) at work is recognized and the importance of exploring new descendants is highly regarded; the current study sheds light on the urge for simultaneous supplementary research on its antecedents too. Relational energy is measured from the receiver’s perspective, as per Owens et al.’s [2] instrument, which is considered to better represent it as opposed to the energy sender’s angle, because the receiver’s approach explains more soundly how the energizing process operates.

To the authors’ knowledge, no other works before have determined this exact correlation among all the variables of the current model. Contribution is also made in intensifying the connection between POS and POB since a combination of variables from both streams is investigated. Furthermore, to the authors’ knowledge, it is the first time this field has been researched in Kosova or the Western Balkans region.

Taking into account all the above, the current work contributes to the expansion of organization and management literature, specifically related to human and organizational energy, positivity, humor, interpersonal communication, social and psychological capital, job engagement, individual and organizational performance, employee wellbeing, healthy work relationships, and motivation.

Regardless of its significant contributions, this work inevitably bears a number of limitations that can represent avenues for future research. The focus on the private service sector only, although including numerous industries within, might miss other important features of interpersonal communication between coworkers in other sectors. Hence, this study may be limited in its generalizability. Other areas, for example, the public sector or manufacturing, could be potential sites for future research that should generalize beyond this work. The concentration on the Prishtina district might be too small to conclude representatively for other cultures and geographies. Irrespective of McDaniel’s [16] findings of no significant cultural differences related to relational energy and that it can be seen as a universal phenomenon, Weng et al. [70] find a stronger crossover of work passion to followers from Anglo cultures than to those from Confucian culture, a relationship that is mediated by relational energy. Moreover, there are other variables that can be more culturally sensitive, such as humor. Therefore, future research that might use the current study’s model can examine it in samples representing other cultures or parts of the world. In order to achieve a model fit, some items needed to be removed from the original scales, especially in the humor case which had several reverse items. One explanation could be that the reading culture in the sample region is such that people prefer short and simple reading. Future research should take this into consideration when designing the questionnaire if they are to carry out research with the same model in cultures with such or similar reading habits. Alternatively, higher-qualified respondents, such as academicians, for example, can be targeted in order to ensure a greater understanding of the questions. Otherwise, in order to shorten the questionnaire and increase the probability of greater focus from the respondents in similar cases, the first and the second half of the model could be examined in separate research studies. The necessity to eliminate a substantial number of items may potentially compromise the current model’s efficacy in generating results that could be widely applied. Consequently, the replication of this study by future research is advantageous. Particularly, the utilization of a more suitable job performance measurement appears critical.

It could be useful for future research to include personality types and/or traits in the analysis. Energy is found to be affected by introversion and extraversion [110], whereas PsyCap as well as humor appeared to be correlated with the Big Five personality traits [111,112].

There might be a likelihood for endogeneity in the sense that, for instance, receivers of relational energy can, in turn, experience growth in their positive humor and PsyCap levels too. As such, future research can replicate the current model by conducting a more meticulous methodological design.

Relational energy can be transmitted from different sources, i.e., coworkers, supervisors, followers, family, friends, and so on. This study’s focus was on coworkers, while many earlier studies investigated merely the leader–follower dyadic interaction. It is advisable for further work to enlarge the relational energy transmission base by a combination of both coworkers and leaders as suppliers of relational energy.

Taking into account that this is a cross-sectional research study, future research shall expand to longitudinal data in order to advance the understanding of the abundance of interactions in a nomological network [76]. Longitudinal studies are especially strongly encouraged since they are expected to be much more reliable in confirming the causal connection and mediation resulting from the current study. Additionally, none of the many control variables significantly correlated with the latent ones. The reason behind this might be that the demographics included do not define the “pepping up” between two people or that other determining control variables are unseen. This could be replicated in the future through reformulation and/or reorganization of some or all current research’s control variables, the inclusion of new ones, or both—reformulation/reorganization and addition.

Albeit there is evidence that the self-assessment scales of job performance and job engagement are not subjective when respondents are assured of no identity disclosure [113]—as is the case in the current study—and that there is no significant difference between self-assessment and other assessments of performance [23]; still, they might fail to capture some objectivity as compared to performance appraisals by supervisor/organization. Prospect studies could increase this accuracy by employing actual performance data such as carried out by Owens et al. [2].

## Figures and Tables

**Figure 1 behavsci-14-00023-f001:**
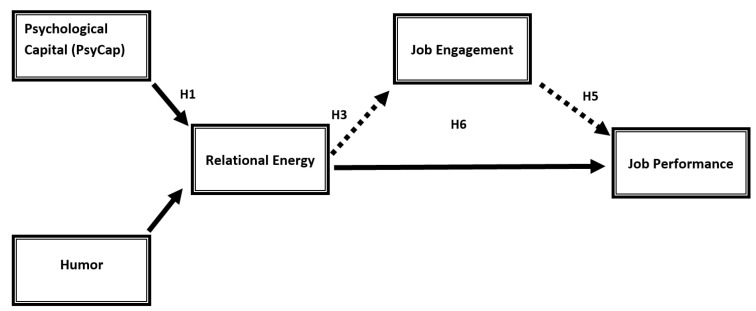
Correlation of relational energy with PsyCap, humor, job engagement, and job performance.

**Figure 2 behavsci-14-00023-f002:**
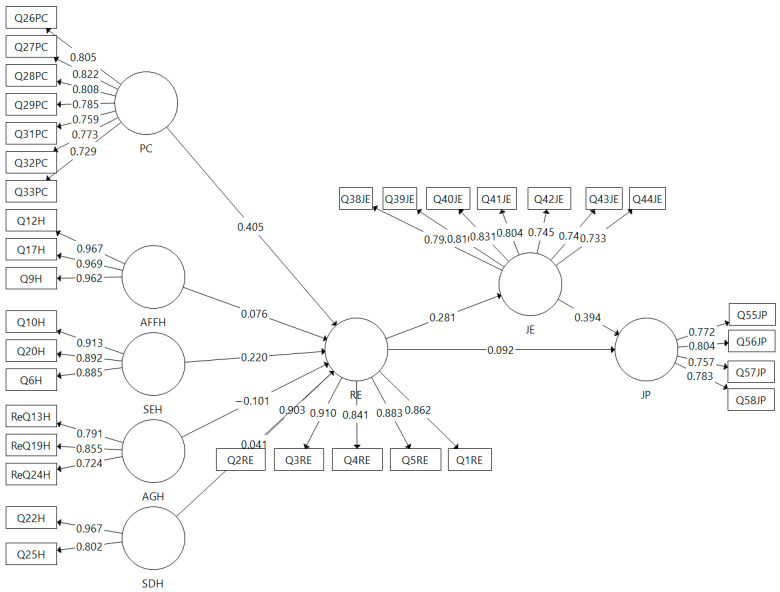
Final model.

**Table 1 behavsci-14-00023-t001:** Mean and standard deviation of the constructs.

Construct	Mean	Standard Deviation
Psychological Capital	4.89	0.88
Affiliated humor	5.17	1.41
Self-enhancing humor	4.89	1.32
Aggressive humor	3.12	1.46
Self-defeating humor	3.72	1.66
Relational energy	5.30	1.22
Job engagement	5.02	0.84
Job performance	4.27	0.63
		*n* (481)

**Table 2 behavsci-14-00023-t002:** Correlation matrix of the constructs.

		PC	AFH	SEH	AGH	SDH	RE	JE	JP
PC	Pearson Correlation	1							
	Sig. (2-tailed)								
AFH	Pearson Correlation	0.291 **	1						
	Sig. (2-tailed)	0.000							
SEH	Pearson Correlation	0.371 **	0.458 **	1					
	Sig. (2-tailed)	0.000	0.000						
AGH	Pearson Correlation	−0.424 **	−0.268 **	−0.380 **	1				
	Sig. (2-tailed)	0.000	0.000	0.000					
SDH	Pearson Correlation	0.004	0.179 **	0.121 **	−0.001	1			
	Sig. (2-tailed)	0.931	0.000	0.008	0.974				
RE	Pearson Correlation	0.551 **	0.332 **	0.452 **	−0.378 **	0.080	1		
	Sig. (2-tailed)	0.000	0.000	0.000	0.000	0.080			
JE	Pearson Correlation	0.300 **	0.116 *	0.157 **	−0.199 **	0.063	0.282 **	1	
	Sig. (2-tailed)	0.000	0.011	0.001	0.000	0.167	0.000		
JP	Pearson Correlation	0.249 **	0.258 **	0.188 **	−0.085	0.025	0.194 **	0.411 **	1
	Sig. (2-tailed)	0.000	0.000	0.000	0.063	0.592	0.000	0.000	

** Correlation is significant at the 0.01 level (2-tailed). * Correlation is significant at the 0.05 level (2-tailed).

**Table 3 behavsci-14-00023-t003:** Reliability analysis of the constructs.

Construct	No. of Items	Cronbach α
Psychological capital	7	0.895
Affiliated humor	3	0.964
Self-enhancing humor	3	0.879
Aggressive humor	3	0.700
Self-defeating humor	2	0.768
Relational energy	5	0.927
Job engagement	7	0.892
Job performance	4	0.785

**Table 4 behavsci-14-00023-t004:** Convergent validity.

Construct	CR	AVE
Psychological capital	0.917	0.614
Affiliated humor	0.977	0.933
Self-enhancing humor	0.925	0.805
Aggressive humor	0.834	0.627
Self-defeating humor	0.881	0.789
Relational energy	0.945	0.775
Job engagement	0.916	0.608
Job performance	0.861	0.607

**Table 5 behavsci-14-00023-t005:** Heterotrait–Monotrait matrix of the constructs.

	PC	AFH	SEH	AGH	SDH	RE	JE
FH	0.319						
SEH	0.421	0.497					
AGH	0.539	0.322	0.485				
SDH	0.037	0.211	0.141	0.150			
RE	0.608	0.351	0.500	0.470	0.096		
JE	0.330	0.125	0.178	0.253	0.079	0.308	
JP	0.300	0.289	0.229	0.126	0.064	0.233	0.496

**Table 6 behavsci-14-00023-t006:** Mediation effect.

Effects	Path	Path Coefficient	Indirect Effect	Total Effect	VAF	t-Value	*p*-Value
Mediator	RE→JE	0.281	Not applicable				
	JE→JP	0.394	Not applicable				
	RE→JP	0.092	0.110	0.203	54.18%	4.583	0.000

Variance accounted for (VAF) = indirect effect/total effect × 100 = (0.110/0.203) × 100 = 54.18%; t-value = indirect effect/standard deviation = 0.110/0.024 = 4.583.

## Data Availability

The data presented in this study are available on request from the corresponding author.

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
