# Peer review of "Energizing Workplace Dynamics: Exploring the Nexus of Relational Energy, Humor, and PsyCap for Enhanced Engagement and Performance"

_behavsci, 2023, doi:10.3390/bs14010023_

Round 1

Reviewer 1 Report

Comments and Suggestions for Authors

Thank you for the opportunity to review this paper. Generally, this paper adopts the empirical research method and conducts SEM test by SmartPLS software. However, the following modifications are still required as it looks like written by a beginner.

1.       It is recommended to write clearly the research gap in the introduction. It would be better to stimulate readers' interest by combing the existing literature. Currently, there is a lack of literature and theoretical foundation for directly proposing research questions.

2.       What is the theoretical foundation of this study?  It’s not clear. The theoretical driving force of the theoretical model are not enough. Why choose psychological capital and humor as driving variables? Why choose job engagement as the moderating variable?

3.       Is there a difference between a circle and a rectangle in the concept model in the figure?

4.       The author does not need to paste each model diagram derived by SmartPLS, but only needs to show the results of the SEM model. Authors are advised to read other high-level/quality papers and refer to their practices.

5.       Write the Conclusions in paragraphs and don't use too many numbers. Please note that this is an academic paper, not a PowerPoint presentation.

 Good luck!

Reviewer 2 Report

Comments and Suggestions for Authors

Thank you for the opportunity to review this paper on the topic of energizing relational dynamics, antecedents, and outcomes. Below are some of my comments, highlighting critical issues in the paper and offering insights for improvement.

1. The introductory section lacks highlights of the problem initiating the study. Why is this issue important in today's world of work? You immediately start with the discussion of the objectives without addressing this part. It is written only, "A broad elaboration of the theoretical illumination supporting the suggested model is to be found in Braha [4,5]." I think this is not enough; it needs to be made explicit and described here as well.

1b. In the article, the topic of antecedents and the consequences of relational energy is discussed first, while the introductory part presents the topics and research questions in reverse order. Unification is needed.

1c. Lines 40-42 lack bibliographical references to the theories.

2. In the section on relational energy, we begin by talking about another construct that has never been mentioned before. Here I expect to read the literature on relational energy and understand what the literature says about it regarding its importance in occupational contexts. Bibliographic references to relevant theories are still lacking.

3. The literature on psycap should be developed a little more by referring, for example, to studies of "collective psycap" or "team-level psycap," but also new research on its outcomes such as safety performance. It is now covered too briefly. For example, refer to:

- Margheritti, S., Negrini, A., & Miglioretti, M. (2023). Can psychological capital promote safety behaviors? A systematic review. International journal of occupational safety and ergonomics, 29(4), 1451-1459.

- Wu, C. M., & Chen, T. J. (2018). Collective psychological capital: Linking shared leadership, organizational commitment, and creativity. International journal of hospitality management, 74, 75-84.

- Newman, A., Ucbasaran, D., Zhu, F. E. I., & Hirst, G. (2014). Psychological capital: A review and synthesis. Journal of organizational behavior, 35(S1), S120-S138.

4. Considering that this is a cross-sectional study, it is not possible to capture the causal link between variables but only their association. Consequently, it is necessary to use different words for "impact" in the formulation of hypotheses.

5. For the same reason, i.e., that this is a cross-sectional study, it would not be possible to assume mediations involving a causal and temporal relationship between variables.

- Maxwell, S. E., Cole, D. A., & Mitchell, M. A. (2011). Bias in cross-sectional analyses of longitudinal mediation: Partial and complete mediation under an autoregressive model. Multivariate behavioral research, 46(5), 816-841.

6. The description of the participants is missing; it is necessary to describe the sample through socio-demographic variables (e.g., gender, age, education, working seniority, working sector, and job title).

7. Line 242: "Subsequently" in what sense? At a distance of time? Is the study longitudinal, or is it cross-sectional with only one questionnaire? This aspect should be made explicit.

8. Line 245-246: The sentence is convoluted; too many reference works are cited, and it is not clear in which work the scale you used is found.

9. Why do you use the label "job engagement" and not "work engagement"? You have referred to the construct defined and studied by Schaufeli, who calls it work engagement. I suggest being more consistent with the referenced literature by using the same terminology.

10. The major criticality of this study lies in the fact that you eliminated several items from the final solution of your SEM model. It is necessary to explain well how you arrived at the reduction of items for the constructs under investigation. Which items were eliminated? This question is critical because by eliminating so many items, the constructs are studied very differently. For example, psycap is composed of 4 personal resources, so you have more items for some dimensions and fewer for others. Similarly, work engagement is composed of three different sub-dimensions (dedication, vigor, and absorption). I am not sure that this model is adequate to test your relationships between variables and arrive at generalizable results.

11. In the results paragraphs, the information is redundant, and the same information is reported many times (e.g., confirmation of H1). It is necessary to revise and streamline this part by eliminating repetitions. It is sufficient to report one overall figure where all coefficients and, in case of indirect effects, are reported in one table.

12. Table 7 is not necessary; what is made explicit in the text is sufficient.

13. I would divide the conclusion paragraph from the practical implications paragraph. Conclusions should be concise, as well as the last paragraph that closes the article by summarizing the key issues. I would rephrase practical implications and limitations not as a bulleted list, and where possible, it is necessary to summarize. These sections can't be longer than the central sections of the study. Some reflection of this kind could be included in the discussions alongside the discussion of whether or not to test the hypotheses.

Round 2

Reviewer 1 Report

Comments and Suggestions for Authors

No more comments.